# Autonomous Exploration Method of Unmanned Ground Vehicles Based on an Incremental B-Spline Probability Roadmap

**DOI:** 10.3390/s24123951

**Published:** 2024-06-18

**Authors:** Xingyang Feng, Hua Cong, Yu Zhang, Mianhao Qiu, Xuesong Hu

**Affiliations:** Army Academy of Armored Forces, Beijing 100072, China; 17695720631@163.com (X.F.); 18911025632@163.com (H.C.); zhangyuzgy2018@163.com (Y.Z.)

**Keywords:** autonomous exploration, unmanned ground vehicles, B-spline curve, probabilistic roadmap

## Abstract

Autonomous exploration in unknown environments is a fundamental problem for the practical application of unmanned ground vehicles (UGVs). However, existing exploration methods face difficulties when directly applied to UGVs due to limited sensory coverage, conservative exploration strategies, inappropriate decision frequencies, and the non-holonomic constraints of wheeled vehicles. In this paper, we present IB-PRM, a hierarchical planning method that combines Incremental B-splines with a probabilistic roadmap, which can support rapid exploration by a UGV in complex unknown environments. We define a new frontier structure that includes both information-gain guidance and a B-spline curve segment with different arrival orientations to satisfy the non-holonomic constraint characteristics of UGVs. We construct and maintain local and global graphs to generate and store filtered frontiers. By jointly solving the Traveling Salesman Problem (TSP) using these frontiers, we obtain the optimal global path traversing feasible frontiers. Finally, we optimize the global path based on the Time Elastic Band (TEB) algorithm to obtain a smooth, continuous, and feasible local trajectory. We conducted comparative experiments with existing advanced exploration methods in simulation environments of different scenarios, and the experimental results demonstrate that our method can effectively improve the efficiency of UGV exploration.

## 1. Introduction

The autonomous exploration of unknown environment map information using unmanned platforms is a fundamental challenge faced by various types of robots such as reconnaissance, rescue, and search. In the selection of different categories of unmanned platforms, existing exploration research often chooses drones or disk-shaped robots with relatively simple kinematic models. However, compared to drones, unmanned ground vehicles (UGVs) have a more detailed modeling capability of ground space due to their rich sensor types. Compared to holonomic-constrained disk robots or Turtlebots, UGVs have stronger maneuverability and passability.

Autonomous exploration based on UGVs aims to plan a continuous, smooth, and vehicle kinematics-constrained motion trajectory to achieve more efficient and comprehensive sensory coverage of unknown environments. Wheeled unmanned ground vehicles have certain special requirements for paths due to their non-holonomic constraint characteristics, and most existing exploration methods are difficult to apply directly. Additionally, limited by the finite sensor perception range and conservative exploration strategies, existing exploration algorithms often exhibit stop-and-go and frequent backtracking phenomena, which incur significant costs for UGVs that need to consider motion orientation. How to achieve feasible, online, and efficient autonomous exploration of UGVs in complex unknown environments is a challenging problem.

This paper addresses the above issues by proposing a hierarchical planning approach that combines Incremental B-splines with a probabilistic roadmap (IB-PRM). We first introduce a new frontier structure, where the frontier is not only a point with information gain but also includes B-spline curve segments with multiple arrival orientations, making it more suitable for the kinematic characteristics of UGVs. Based on this, we store and filter these frontiers by constructing and maintaining global and local graph structures. The local graph is used to generate new frontiers and paths to reach these frontiers within the perception range of the UGV, while the global graph is used to store all generated frontiers and paths to these frontiers. By jointly solving the Traveling Salesman Problem (TSP) on the frontiers, we obtain the global path traversing feasible frontiers, avoiding greedy navigation to maximize immediate information gain or the nearest unknown area and falling into a local optimum. Subsequently, the global path is optimized using the Timed Elastic Band algorithm to derive a local path suitable for execution by the UGV. Finally, we conducted comparisons with mainstream exploration algorithms in different scenarios using the Gazebo simulation platform to demonstrate the superiority of our proposed method. The main contributions of this paper can be summarized as follows:
(1)We propose a hierarchical planning approach for the autonomous exploration of UGVs. By constructing global and local graphs and solving the TSP, we obtain a global planning path suitable for UGVs and use the Time Elastic Band (TEB) algorithm to optimize the global path for generating continuous, smooth local trajectories that meet kinematic constraints. The introduction of the global graph increases the probability of finding better viewpoints and escaping deadlocks, while hierarchical planning enhances the smoothness and stability of vehicle motion.(2)We improved the encoding method for expanding nodes in the PRM search algorithm and proposed a search algorithm that forms a new node by encoding the control points of a cubic B-spline curve, combining four neighboring nodes with connectivity into a new node. This allows the nodes to include high-order continuous curve segments, enhancing the ability of local graphs for path-smoothing optimization.(3)We redefine frontiers suitable for non-holonomic constraint robots. First, select suitable nodes in the PRM based on information gain values, and then cluster the filtered nodes in the local graph according to path similarity as our defined frontiers. The new frontiers not only serve as guides for information gain but also represent a segment of smooth and continuous cubic B-spline curves, helping the robot explore more efficiently and continuously.

## 2. Related Work

Using robots for the construction and exploration of unknown environments has a long research history, essentially falling under the category of Active Simultaneous Localization and Mapping (Active-SLAM) [1]. Various proposed methods, based on the selection of exploration objectives, can generally be divided into frontier-based methods and the Next Best View method.

Frontier-based exploration refers to guiding robot movements by detecting the boundaries between known and unknown areas on the map. The concept of frontiers was introduced in [2], where edge detection techniques in computer vision were used to extract cells adjacent to unknown spaces as frontiers and consider the nearest frontier as the robot’s target. However, as the exploration area expands, it leads to the curse of dimensionality in detection. Umari et al. [3] improved this by establishing a double-layer expandable Rapidly Exploring Random Tree (RRT) to define nodes with a higher proportion of unknown areas within the perception range as frontiers. Using the expansiveness and probabilistic completeness of RRT sampling, the local RRT tree increases the probability of the robot sampling frontiers in unknown space, while the global RRT ensures the integrity of map exploration. However, due to the lack of consideration for the robot’s kinematic constraints in the selection of target frontiers, it is difficult to apply to unmanned ground vehicle exploration. In [4], a differentiable gain based on the frontiers was introduced, allowing the optimization of paths using gradient information to guide robot movement, enhancing trajectory continuity. Zhou et al. [5] proposed an incrementally updated frontier information structure (FIS), which sampled viewpoints with high frontier coverage to reduce the number of candidate frontiers. Finally, by solving the Asymmetric Traveling Salesman Problem (ATSP), they obtained a suboptimal global path traversing the current frontiers. This method was further expanded in [6] and applied to collaborative exploration by multiple drones, demonstrating high efficiency. The objective of the exploration method based on frontiers is completely decoupled from motion planning, leading to the need to replan paths for all candidate positions during exploration replanning, which consumes a considerable amount of computational resources and may lead to frequent stops and reversals.

The essence of the Next Best View (NBV) is a variant of sensor placement problems [7]. González-Baños et al. [8] sampled candidate viewpoint location sets in the largest obstacle-free region around the robot, selecting the viewpoint location with maximum expected gains to guide the robot’s movement and construct the environmental map. This method balances the cost of moving to the next viewpoint and the exploration gain obtained at that location, enabling the robot to explore a larger map area in the early stages of exploration. Building on this, Bircher et al. [9] proposed an NBV algorithm based on receding horizon (RH-NBV), considering the cost of the RRT path and the gain value of nodes, selecting a branch with the highest combined return as the optimal branch. The robot only executes the first edge of this branch, initializing the remaining part of the tree. However, due to the lack of global considerations, it may fall into local optima. Witting et al. [10] improved the sampling-based NBV exploration in terms of sampling direction and trajectory generation, introducing a history graph to store distributed information of explored space to avoid dead ends. This paper is also inspired by the introduction of the global–local bilayer graph structure. Respall et al. [11] also introduced a similar historical graph for quickly finding potential exploration areas. Dang [12] combined frontier-based and NBV-based algorithms, presenting a topology-based autonomous exploration strategy (GB) by clustering paths in the local graph, and adding them to the global PRM graph to avoid prematurely ending the exploration. Selin et al. [13] combined RH-NBV and frontier-based methods, proposing the Autonomous Exploration Planner (AEP). Utilizing frontiers for global path planning and NBV algorithms for local path planning, it overcomes the slow exploration issues of frontier theory in three-dimensional environments, while avoiding local optima that NBV algorithms may bring. The exploration algorithm based on the NBV concept iteratively searches for viewpoints with the richest perceptual information. However, this strategy may reduce the robot’s attention to areas with less information gain, causing it to ignore these small regions during the exploration process, potentially leading it to fall into local minima.

Overall, most existing methods tend to make decisions greedily and do not consider the kinematic constraints of wheeled vehicles, resulting in inefficient global trips and conservative maneuverability. In contrast, we plan to effectively cover the entire environment and generate dynamically feasible optimal trajectories to achieve continuous efficient motion.

## 3. Background

The mathematical description of the autonomous exploration problem and the kinematic model of the vehicle form the foundation for the subsequent research. Therefore, we will first present this in this section for further elaboration and comprehension.

### 3.1. Autonomous Exploration Problem Description

Let G represent the entire map that needs to be explored and M represent the current saved grid map, which is gradually updated under the action of sensors. As the vehicle moves, with the current position ξ(x,y,β) of the vehicle as the center, the circular sensor perception area with a radius $r_{sensor}$ updates the map state. Here, the map is divided into free, obstacle, and unknown areas, corresponding to symbols $M_{\mathrm{free}}$, $M_{\mathrm{occupied}}$, and $M_{\mathrm{unknown}}$, respectively. Assuming the vehicle reaches a certain point $P_{target}$, establishing a perception area S with that point as the center, the area in S where the internal grid changes from unknown to obstacle or free is taken as the information gain value $gain_{P}$ of point $P_{target}$. As shown in Figure 1, the white area represents free space, the black area represents grids occupied by obstacles, and the gray area represents unknown regions. The black dashed circle represents the perception area S, and in the actual calculation process, we use a ray casting algorithm to estimate the information gain value of that point. The exploration process of the robot can be roughly described as the robot continuously updating target point $P_{target}$ and exploration path τ based on real-time environmental information and utilizing sensors to achieve complete coverage of unknown environments while executing path $\tau^{*}$ (set of τ). However, during the vehicle exploration process, due to constraints such as its kinematic constraints, there are areas in the map that the perception module cannot observe, such as closed small spaces or obscured holes, resulting in certain residual unknown areas $M_{*,res}\subset M_{\mathrm{unknown}}$. Therefore, we define the concept of region exploration completion as follows: for any point $P(x,y),P\in M_{\mathrm{free}}$ in the current region, the information gain value $gain_{p}<\varepsilon$, where ε is a small quantity, then the region exploration is completed. When any point in the entire map meets the above condition, the global exploration is completed. The main parameters used in this paper are listed in Appendix A.

### 3.2. Vehicle Kinematics Model

The selection of a suitable kinematic model for autonomous vehicles is the basis for subsequent autonomous planning. Considering that vehicles, especially during autonomous exploration and turning processes, have low speeds, the lateral tire movements are ignored, allowing the vehicle’s tires to satisfy non-holonomic constraints relative to the ground. This results in a simplified “bicycle model”, which meets the requirements while avoiding the computational complexity of more elaborate models. As shown in Figure 2, with the vehicle’s body axis distance as L and the wheelbase as $2l_{r}$, the vehicle’s state variables are denoted as ξ(x,y,β), where x and y represent the vehicle’s Cartesian coordinates, β is the vehicle’s traverse angle, ψ is the vehicle’s center-of-mass lateral deflection angle, v is the vehicle’s velocity, and δ is the equivalent front-wheel steering angle. They are all functions of time t, leading to the following kinematic equations for the vehicle:(1)$$\xi \left(t\right)=\left[\begin{array}{c} \dot{x}\left(t\right)\\ \dot{y}\left(t\right)\\ \dot{\beta }\left(t\right) \end{array}\right]=\left[\begin{array}{c} v\left(t\right)\cos\left(\psi \left(t\right)+\beta \left(t\right)\right)\\ v\left(t\right)\sin\left(\psi \left(t\right)+\beta \left(t\right)\right)\\ v\left(t\right)\frac{\tan\left(\delta \left(t\right)\right)}{L}\cos\left(\psi \left(t\right)\right) \end{array}\right]$$

## 4. Proposed Approach

Correct target point guidance and updating are key to efficient exploration. As introduced in the background in Section 3, as the exploration progresses, the posterior distribution of environmental information will change, causing the frontiers chosen in the previous moments to shift gradually from optimal to suboptimal. A low frequency of selecting target points can lead to vehicles heading to areas of the map that have already been updated, inefficiently following unnecessary paths. Conversely, a high frequency can make it difficult for the robot to find a stable, long-term executing target, resulting in the robot oscillating back and forth in the region [14]. Therefore, improving the efficiency and continuity of exploration while satisfying the constraints of vehicle kinematics and obstacle avoidance is our main concern.

We achieve our goal by proposing a hierarchical planning method based on IB-PRM. Figure 3 illustrates the system framework of the method proposed by us. Whenever the unmanned ground vehicle reaches a new location, a PRM graph is randomly sampled within the perception range, with the current position as the center. Then, using the cubic B-spline control point search algorithm, the shortest path from the starting node to each node in the graph is obtained (corresponding to the green dashed line part in Figure 3, Section 4.1), and a new local graph structure is established (corresponding to the red dashed line part in Figure 3, Section 4.2). After clustering the nodes in the local graph, they are combined as new frontiers with the historical frontiers already stored in the global graph to solve the Traveling Salesman Problem and obtain the globally optimal path traversing the best nodes (corresponding to the blue dashed line part in Figure 3, Section 4.3). Meanwhile, the frontiers in the local graph will also be added to the global graph in the form of paths. Finally, the TEB algorithm is used to generate a locally smooth path suitable for the unmanned ground vehicle to execute (corresponding to the yellow dashed line part in Figure 3, Section 4.4), which is then output to the controller to complete one exploration motion. Iterate the process until the conditions for completing the full-graph exploration as set by us are met. The B-spline control point search algorithm proposed by us, which serves as the foundation and main innovation point for the subsequent content, will be first introduced in Section 4.1. The design and theoretical derivation of the local trajectory planner will be detailed in Section 4.4.

### 4.1. B-Spline Control Points Search

B-splines possess excellent properties such as locality, continuity, convex hull characteristics, and geometric invariance. Building upon the B-spline search [15,16], we propose a B-spline control point search algorithm utilizing a probabilistic roadmap. On one hand, most planning algorithms adopt a hierarchical framework, utilizing low-dimensional global planning paths as references and incorporating time dimensions using numerical optimization methods while satisfying motion and collision constraints. The exploration process of autonomous vehicles similarly follows this approach. However, the exploration process lacks fixed target points; its primary objective is to guide the vehicle to unknown areas. In this scenario, a fast and feasible path search algorithm is needed to acquire paths to reach frontiers. On the other hand, with map updates, the exploration target points of the vehicle change, leading to corresponding changes in global paths. Frequent changes in global paths can cause discontinuous vehicle motion, prolonging exploration time. Hence, an algorithm considering the current orientation of the vehicle is essential to ensure continuity throughout the global planning. We have designed a B-spline control point search algorithm based on PRM to meet the aforementioned requirements. In comparison to traditional search algorithms, using PRM to represent the map significantly reduces the number of nodes while ensuring probabilistic completeness. Expanding neighboring nodes through a B-spline search leads to a high-order continuous global path.

Let $\pi =(P_{0},\;P_{1},\ldots ,\;P_{T})$ be the control point representing the optimal trajectory. According to the locality of B-spline curves, for a *k*-degree B-spline curve, every k+1 control point determines a segment of the curve, denoted as $\pi_{i}=(P_{0},\;P_{1},\ldots ,\;P_{k})$. To accurately quantify the cost of this segment, an appropriate cost function is needed. Mellinger and Kumar [17] proposed that the quadratic integral of trajectory derivatives can reflect the smoothness of the trajectory and the cost of control to some extent, but calculating high-order derivatives of the trajectory is complex. Therefore, Usenko [18] used the control points of the B-spline to calculate the integrals of the required high-order derivatives of the trajectory and optimized them as the objective function of the curve. The cost function of a cubic B-spline curve is as follows:(2)$$E_{q}{|}_{k=3}={\left[\begin{array}{c} P_{i-3}\\ P_{i-2}\\ P_{i-1}\\ P_{i} \end{array}\right]}^{T}M^{T}QM\left[\begin{array}{c} P_{i-3}\\ P_{i-2}\\ P_{i-1}\\ P_{i} \end{array}\right],$$
where M represents the basis function matrix of the curve, and Q represents the integration of the interval vector’s derivatives. Their expressions are as follows:(3)$$\left\{\begin{array}{c} M=\frac{1}{3!}\left[\begin{array}{cccc} 1 & 4 & 1 & 0\\ -3 & 0 & 3 & 0\\ 3 & -6 & 3 & 0\\ -1 & 3 & -3 & 1 \end{array}\right]\\ Q=\frac{1}{\Delta t^{3}}\int_{0}^{1}\left[\begin{array}{c} 0\\ 0\\ 2\\ 6u(t) \end{array}\right]{\left[\begin{array}{c} 0\\ 0\\ 2\\ 6u(t) \end{array}\right]}^{T}dt \end{array}\right.,$$
where Δt represents the interval of the B-spline parameter *t*, u(t) represents the normalized result of the original B-spline parameter equation, i.e., $u(t)=(t-t_{i})/(t_{i+1}-t_{i})$.

From this, it can be seen that the cost of the curve $c(t),t\in [t_{i},t_{i+1})$ is not only controlled by two simple starting and ending points, but collectively determined by the k+1 control points $\pi_{i}=(P_{0},\;P_{1},\ldots ,\;P_{k})$. In this case, we define these k+1 control points as nodes $\tilde{P}$ of a new graph $G_{H}<\tilde{P},\tilde{E}>$, where edges $\tilde{E}$ represent the connectivity between the new node ${\tilde{P}}_{i}$. Figure 4 simply depicts the correspondence between the graphs $G_{H}<\tilde{P},\tilde{E}>$ and the original graph G<V,E> when k=3. The nodes $P_{i}(i=0,1,2,3,\ldots ,n)$ in the graph form an initial graph structure G<P,E>, with E representing the edges between nodes. Based on this, the graph $G_{H}$ is constructed with 4 nodes as a new node ${\tilde{P}}_{i}$, where $P_{0}$, $P_{1}$, $P_{2}$, and $P_{3}$ form a new node ${\tilde{P}}_{0}$, $P_{1}$, $P_{2}$, $P_{3}$, and $P_{4}$ form a new node ${\tilde{P}}_{1}$. Since $P_{3}$ is connected to $P_{4}$ in graph G, ${\tilde{P}}_{1}$ and ${\tilde{P}}_{0}$ share $P_{1}$, $P_{2}$, and $P_{3}$, ${\tilde{P}}_{1}$ is connected to ${\tilde{P}}_{0}$, and, similarly, ${\tilde{P}}_{0}$ is connected to ${\tilde{P}}_{2}$. As a result, the new node $\tilde{P}$ jointly defined by the four original nodes *P* represents a segment of a high-order continuous curve in geometric terms, thereby transforming the original path $\pi =(P_{0},\;P_{1},\ldots ,\;P_{T})$ into a new set of paths $\pi =({\tilde{P}}_{0},\;{\tilde{P}}_{1},\ldots ,\;{\tilde{P}}_{T})$.

Next, we draw on the idea of the Dijkstra algorithm. First, we randomly sample points in the map where the distance to obstacles is greater than $d_{check}$ to generate the probabilistic roadmap G. Subsequently, we use B-spline control points to search the entire region’s graph structure, finding the shortest path from the starting point to each node in the graph. The algorithm flow is shown in Algorithm 1. Initially, we maintain three container structures: *OPEN*, *LIST*, and *CLOSE*. *OPEN* and *LIST* structures help in finding the node ${\hat{v}}_{i}$ with the minimum cost value, while *CLOSE* determines if the node has already found the minimum value for the path. We calculate the cost value $g({\tilde{P}}_{s})$ of the starting node using Equation (2). To differentiate between the expanded nodes, we use the function INDEX(⋅) to encode each node. Considering that the orientation of the initial node of each segment of a cubic spline curve is controlled only by the first and third nodes, we generate an *index* value for each new node ${\tilde{P}}_{i}=(P_{j},P_{j+1},P_{j+2},P_{j+3})$ based on encoding the first node $P_{j}$ and the third node $P_{j+2}$ among the four nodes.

We use the function CurveExpand(⋅) to expand the next node while checking the feasibility of the node. First, we obtain the tail node $\;P_{k+i}$ of the current node $m=(P_{i},P_{i+1},\ldots ,\;P_{k+i})$ in the original PRM graph. By using the connectivity in the PRM graph, we obtain its neighboring node $P_{j}$ and combine node $P_{j}$ with the three nodes $P_{k+i}$, $P_{k+i-1}$, and $P_{k+i-2}$ in $m=(P_{i},P_{i+1},\ldots ,\;P_{k+i})$ to form a new node ${\hat{v}}_{j}$. To reduce computational complexity, the feasibility of the node is checked by simply detecting that there is no collision between the control points. In the absence of a target point, the entire search process continues until the OPEN set is empty, thus obtaining the shortest paths from the starting point ${\tilde{P}}_{s}$ to each point ${\tilde{P}}_{i}$. After obtaining the shortest paths, precise collision detection is performed on the curve interpolation. We adopt the method from [19], by inserting new control points on the line connecting the control points corresponding to the collision position to increase the weight in the direction away from obstacles, ultimately obtaining a safe collision-free path.
**Algorithm 1:** B-spline Node Search**1:****function** GetBsplineShortPath **(**${\tilde{P}}_{s}$*,k*,Δt,G**)****1:****function** GetBsplineShortPath **(**${\tilde{P}}_{s}$*,k*,Δt,G**)****2:**  OPEN←∅;CLOSE←∅;LIST←∅**//init priority queue****3:**  
$g({\tilde{P}}_{s})\leftarrow E_{q}{|}_{k,\Delta t}({\hat{v}}_{j})$
**4:**  
$OPEN\leftarrow INSERT(OPEN,g({\tilde{P}}_{s}),{\tilde{P}}_{s})$
**5:**  
$m\leftarrow INDEX({\tilde{P}}_{s})$
**6:**  
$LIST\leftarrow INSERT(LIST,m,{\tilde{P}}_{s})$
**7:**  **while** OPEN≠∅ **do****8:**    
$\left(m,{\hat{v}}_{i})\leftarrow POP(OPEN\right)$
**9:**    **for** ${\hat{v}}_{j}\in CurveExpand({\hat{v}}_{i},G,k,\Delta t)$ **do****10:**      
$m\leftarrow INDEX({\hat{v}}_{j})$
**11:**      **if not** VISITED(n,LIST) **then****12:**        
$g({\hat{v}}_{j})\leftarrow \infty$
**13:**      
**end if**
**14:**      **if** $g({\hat{v}}_{j})>g({\hat{v}}_{i})+E_{q}{|}_{k,\Delta t}({\hat{v}}_{j})$ **then****15:**        
$g({\hat{v}}_{j})=g({\hat{v}}_{i})+E_{q}{|}_{k,\Delta t}({\hat{v}}_{j})$
**16:**        
$OPEN\leftarrow INSERT(OPEN,g({\hat{v}}_{j}),{\hat{v}}_{j})$
**17:**      
**end if**
**18:**    
**end for**
**19:**  
**end while**
**20:****end function**

### 4.2. Local Graph Construction

At the beginning of the exploration, the global exploration graph is initialized using the starting node. Then, the current pose $\xi_{0}(x_{0},y_{0},\beta_{0})$ of the vehicle is obtained. Taking the current position of the vehicle as the center and the perception length $r_{sensor}$ as the radius, a local PRM is constructed. As shown in Algorithm 2, new nodes are sampled within the perception range S. Subsequently, using *KD-TREE*, the nearest node in the graph is found, and nodes are added at a distance of step size $d_{grow}$. Collision detection is performed using lazy check. Connection edges are formed between collision-free sampling nodes and neighboring nodes, and it is determined whether there are connectable edges within a distance of $d_{near}$ around the new node. The construction of the local PRM graph G<V,E> continues until the number of nodes or edges in the entire graph exceeds the set thresholds $Node_{max}$ and $Eage_{max}$.
**Algorithm 2:** Build Local PRM**1:****function** *BuildLocalGraph* **(**$\xi_{0}$ **)****2:**  
$V\leftarrow \xi_{0};E\leftarrow \emptyset$
**3:**  
$G_{L}=(V,E);D_{L}\leftarrow SetLocalBound(\xi_{0})$
**4:**  **while** $N_{V}<N_{V,max}$ **and** $N_{E}<N_{E,max}$ **do****5:**    
$\xi_{rand}\leftarrow SampleFree(S_{L})$
**6:**    
$\xi_{nearest}\leftarrow NearestVertex(G_{L},\;\xi_{rand})$
**7:**    **if CollisionFree**$\left(\xi_{rand},\;\xi_{nearest}\right)$ **then****8:**      
$V\leftarrow V\cup \{\xi_{rand}\};E\leftarrow E\cup \{\xi_{rand},\;\xi_{nearest}\}$
**9:**      
$V_{near}\leftarrow NearestVertices(G_{L},\;\xi_{rand},\;\delta )$
**10:**      **for all** $V_{near}\in V_{near}$ **do****11:**        **if CollisionFree**$\left((\xi_{rand},\;V_{nearest})\right)$ **then****12:**          
$E\leftarrow E\cup \{\xi_{rand},\;\xi_{near}\}$
**13:**        
**end if**
**14:**      
**end for**
**15:**    
**end if**
**16:**  
**end while**
**17:**  **return** $G_{L}=(V,E);$**18:****end function**

Subsequently, the cubic B-spline control point search algorithm in Algorithm 1 is used to obtain the optimal path from the starting point to each node. The B-spline graph search algorithm, based on graph G<V,E>, combines 4 adjacent nodes $P_{i}$, $P_{i+1}$, $P_{i+2}$, and $P_{i+3}$ to form a new node ${\tilde{P}}_{j}$ and stores it in the tail node $P_{i+3}$ to establish a new local graph $G_{H}<\tilde{V},\tilde{E}>$. This combination ensures that even if the tail node $P_{i+3}$ is the same among the four nodes, i.e., even if reaching the same node $P_{i+3}$ during the expansion process, the different expansion paths result in a different *index* for the new node $\tilde{P}$. After the search, each node $P_{i}$ in $G_{H}<\tilde{V},\tilde{E}>$ stores a large number of new nodes ${\tilde{P}}_{j}$. Based on this special graph structure, suitable frontier sets are selected through two layers of operations: filtering and clustering.
(1)Filtering: The primary goal of exploration is to expand the unknown areas on the map. Therefore, we use the ray-casting algorithm to calculate the information gain value gain of nodes in the local graph for filtering. Unlike algorithms [9,12] where the node with the highest information gain value is selected as the target point, we do not rank the information gain values of nodes. Considering that the vehicle’s perception range is a circular area with a radius of $r_{sensor}$, the orientation of nodes during exploration does not affect the magnitude of information gain value. The calculation of information gain value is solely for filtering out frontiers and non-frontiers. We set a threshold $gain_{threshold}$ as a boundary, where points $gain>gain_{threshold}$ are considered frontiers, completing the first filtering step.(2)Clustering: After filtering, there are still many frontiers in the area, and many of these frontiers are close in position and orientation. Directly merging these frontiers with frontiers in the global graph for solving the Traveling Salesman Problem undoubtedly increases computational complexity. Therefore, we use the Dynamic Time Warping algorithm (DTW) [20] to cluster the remaining nodes. This algorithm compares the similarity between different paths and cluster nodes with similar and close positions based on the temporal distance between paths. The number of frontiers significantly reduces after DTW clustering.

### 4.3. Global Graph Construction and TSP Solution

To preserve the generated frontiers, we establish and maintain a global graph structure that is not reinitialized with each iteration of the exploration algorithm but incrementally expands as the map is updated. During the expansion process, new frontiers generated in the local graph are added to the global graph in a path-like manner using a method similar to Algorithm 2. This ensures connectivity and allows the utilization of the global graph to find frontiers further away from the vehicle, providing retreat routes when the vehicle encounters dead ends in local areas. Additionally, the gradually constructed global graph only incorporates paths selected from the local graph, reducing the search burden and laying the groundwork for handling large, complex environments.

The Traveling Salesman Problem involves finding the shortest path for a single traveler to depart from a starting point, visit all target nodes, and return to the origin. The purpose of exploration is to quickly traverse the entire graph. In contrast to algorithms [9] and [12], where only the node with the lowest local cost is selected as the target point, leading to potential local optima, a comprehensive consideration of all node paths results in shorter and more effective exploration paths. In each frame of the algorithm loop, multiple frontiers are generated. Inspired by [5], this paper converts these into a Traveling Salesman Problem, incorporating improvements based on the characteristics of unmanned vehicles.

When new frontiers are generated in the local graph, after filtering and clustering, combined with existing frontiers in the global graph, the traversal problem of frontiers is transformed into a Traveling Salesman Problem for resolution. When the information gain value gain for each node in the local area is less than a threshold $gain_{threshold}$, indicating the completion of the local area exploration, only historical frontiers stored in the global graph are used to solve the Traveling Salesman Problem to obtain escape routes.

Assuming the existence of N frontiers denoted as $F_{i}(i=1,\cdots ,N)$, where $F_{0}$ represents the node where the vehicle is currently located, the starting point of the TSP. In existing algorithms, drones, due to their few constraints, adopt an approximation of circular flying during exploration. While this method reduces exploration path length and time, it is not suitable for vehicles with more constraints. Additionally, cyclically solving the TSP and executing it consumes significant computational resources during exploration. As robots can only move to one node at a time during each TSP loop, the computation cost of calculating the true path cost for all nodes is high. Therefore, considering the relatively low movement cost of nodes in the local graph, prioritizing the newly generated frontiers in the local PRM is likely to yield better results. In the calculation of the cost from the starting point to each frontier, the cost from the starting point $F_{0}$ to frontiers outside the local graph is set to a large value, while the cost between other nodes is calculated using the normal formula. The formula is as follows:(4)$$c_{ij}=\max\left\{length(dijistra(F_{i},F_{j})),length(Bspline(F_{i},F_{j}))\right\}.$$

In the equation, $c_{ij}$ represents the cost from frontier $F_{i}$ to frontier $F_{j}$, which is the maximum value of the path length obtained from the Dijkstra search considering obstacle avoidance constraints and the B-spline curve length ignoring obstacles. The design of the equation is inspired by the design concept of the hybrid A* heuristic function [21], utilizing the search speed of the Dijkstra algorithm to obtain the collision-free shortest path for each node. Combining the locality and continuity of the B-spline curve (ignoring obstacles) to find a smooth trajectory suitable for unmanned vehicle execution, the maximum value between the two is taken as the cost between nodes, avoiding the excessive computational cost of considering both kinematic and obstacle avoidance constraints for each node. The cost of the B-spline curve between nodes is directly retrieved from the global graph without the need for separate solving.

Considering that the TSP requires returning to the starting point, while exploration does not require such a return, this paper sets the distance from each frontier to the starting point $F_{0}$ as 0. After solving the TSP, the order of the shortest paths for traversing each node is obtained, with the first node to be traversed selected as the current target point. Additionally, considering the uncertainty in the vehicle’s movement process due to modules such as localization leads to ongoing oscillation in the vehicle’s selection of target points, preventing the stable movement towards the target point and the rolling optimization of the arrival path. We only call the LKH library [22] to solve the TSP when the change in information gain value at the target frontier is less than a certain threshold $\Delta gain_{TSP}$.

### 4.4. Local Path Planning Based on TEB

To ensure the quick identification of frontiers in the map, we adopt a strategy of non-uniform step extension, which increases the density of the graph, resulting in unequal distances between nodes in the path. When the distance is large, the local trajectory appears rough and it is difficult to guarantee optimality. Therefore, trajectory optimization is required for the global path that includes B-spline curve segments obtained from the search. The TEB algorithm [23] divides the trajectory into segments consisting of line segments and circular arcs, capable of generating the optimal executable trajectory while satisfying obstacle avoidance and dynamic constraints.

The traditional Elastic Band (EB) algorithm uses the vehicle’s pose s(x,y,β) for path optimization and defines the entire optimization variables as follows:(5)$$Q={\left\{s_{i}\right\}}_{i=0\ldots n}$$

TEB then adds the corresponding time interval ΔT on the basis of EB, with the formula as follows:(6)$$\tau =\{\Delta T_{i}{\}}_{i=0\ldots n-1},$$
where $\Delta T_{i}$ represents the time needed to transform posture $s_{i}(x,y,\beta )$ to posture $s_{i+1}(x,y,\beta )$, and combining the two equations gives the optimization variable of TEB as follows:(7)$$B:=(Q,\tau ):=\{s_{1},\Delta T_{1},s_{2},\Delta T_{2},\ldots ,s_{n-1},\Delta T_{n-1},s_{n}\}$$

As shown in Figure 5a, $s_{i}(x_{i},y_{i},\beta_{i})$, $s_{i+1}(x_{i+1},y_{i+1},\beta_{i+1})$, and $s_{i+2}(x_{i+2},y_{i+2},\beta_{i+2})$ are three poses on the time elastic band, with $\Delta T_{i}$ and $\Delta T_{i+1}$ representing the time intervals between $s_{i}$ and $s_{i+1}$ and $s_{i+1}$ and $s_{i+2}$, respectively. The principle is to discretize the entire path, treating the entire path as an elastic band, with path points serving as nodes of the elastic band. The constraints and objective function of the path are considered as external forces of the elastic band, and appropriate paths are found by modifying nodes based on external forces. This algorithm combines local planning with control quantities, taking the global path as input and motor speed and wheel angle as outputs. Interpolation is performed on the basis of the global path $\sigma_{best}=[\xi_{0},\xi_{1},\xi_{2},\ldots ,\xi_{n}]$ to obtain an initial path $s_{best}=[s_{0},s_{1},s_{2},\ldots ,s_{n}]$, where both ξ(x,y,β) and s(x,y,β) represent the pose of the vehicle.

Given the trajectory of a path, combined with Equation (1), the motor speed $v_{control}$ and the equivalent front wheel steering angle $\delta_{control}$ at each moment t can be derived as follows:(8)$$u\left(t\right)=\left[\begin{array}{c} v_{control}\left(t\right)\\ \delta_{control}\left(t\right) \end{array}\right]=\left[\begin{array}{c} \gamma \left(\cdot \right)\sqrt{\dot{x}{\left(t\right)}^{2}+\dot{y}{\left(t\right)}^{2}}\\ \tan^{-1}L\frac{\dot{\beta }\left(t\right)}{v\left(t\right)\cos(\psi \left(t\right))} \end{array}\right]$$
where $\gamma \left(\cdot \right)\in \left[-1,1\right]$ indicates the direction of linear velocity, which is positive for forward and negative for backward.

When planning the local trajectory, we need to consider various constraints such as the global path, obstacles, vehicle kinematics, and dynamics. Firstly, there are kinematic constraints. As shown in Figure 5b, let $s_{k}$ and $s_{k+1}$ be two consecutive poses in the vehicle trajectory, $x_{k}$ and $x_{k+1}$ be the velocity directions of the two poses, $d_{k}=\left[x_{k+1}-x_{k},\;y_{k+1}-y_{k}\right]$, $v_{k,k}$ and $v_{k,k+1}$ be the angles between $x_{k}$ and $d_{k}$ and $x_{k+1}$ and $d_{k}$, respectively. Since the two poses are on the same constant curvature arc with curvature $\rho_{k}$, the vehicle moves from $s_{k}$ to $s_{k+1}$ only under the action of a constant control quantity $u_{k}$. Combining Equation (8) leads to the derivation of the vehicle kinematic constraints.
(9)$$h_{k}\left(s_{k+1},s_{k}\right)=\left(\left[\begin{array}{c} cos\left(\beta_{k}\right)\\ sin\left(\beta_{k}\right)\\ 0 \end{array}\right]\right)+\left(\left[\begin{array}{c} cos\left(\beta_{k+1}\right)\\ sin\left(\beta_{k+1}\right)\\ 0 \end{array}\right]\right)\times d_{k}=0$$

In addition, the constraints of maximum curvature, maximum velocity, and maximum acceleration are, respectively:(10)$$\rho_{k}=\frac{{\left\|d_{k}\right\|}_{2}}{\left|2sin(\frac{\Delta \beta_{k}}{2})\right|}\overset{\Delta \beta_{k}\ll 1}{\approx }\frac{{\left\|d_{k}\right\|}_{2}}{\left|\Delta \beta_{k}\right|}<\rho_{max}$$
(11)$$\upsilon (s_{k+1},s_{k},\Delta T_{k})={\left[v_{max}-\left|v_{k}\right|,\omega_{max}-\left|\omega_{k}\right|\right]}^{T}\geq 0$$
(12)$$\alpha_{k}\left(s_{k+2},\;s_{k+1},\;s_{k},\;\Delta T_{k+1},\;\Delta T_{k}\right)=a_{max}-\left|a_{k}\right|\geq 0,$$
where $v_{k}\approx \gamma {\left\|d_{k}\right\|}_{2}\left(s_{k},s_{k+1}\right)/\Delta T_{k}$, angular velocity $\omega_{k}=\Delta \beta_{k}/\Delta T_{k}$, and acceleration $a_{k}=2(v_{k+1}-v_{k})/(\Delta T_{k}+\Delta T_{k+1})$.

For obstacle constraints, we utilize the discrete nature of trajectory points to consider obstacle avoidance by taking into account the poses near obstacles. Firstly, find the trajectory points in the entire global path within a certain range of obstacles $O_{k}$ and their corresponding obstacles. Utilize the scalar $\varepsilon_{min}$ to constrain their distances, with the constraint inequality as follows:(13)$$o_{k}(s_{k})={\left[\delta \left(s_{1},O_{1}\right),\delta \left(s_{2},O_{2}\right),\ldots ,\delta \left(s_{k},O_{k}\right)\right]}^{T}-{\left[\varepsilon_{min},\varepsilon_{min},\ldots ,\varepsilon_{min}\right]}^{T}\geq 0,$$
where $\delta \left(s_{k},O_{k}\right)$ represents the minimum distance from pose $s_{k}$ to obstacle $O_{k}$, and scalar $\varepsilon_{min}$ represents the minimum distance that can be operated with the obstacle.

At the initial stage, the TEB will be initialized using the global planning path, and as the vehicle moves, the optimized state variables will be dynamically removed and added, with the adjustment intervals as follows:(14)$$\Delta T_{ref}-\Delta T_{hyst}<\Delta T_{k}<\Delta T_{ref}+\Delta T_{hyst},$$
the $\Delta T_{ref}$ represents the time interval that $\Delta T_{k}$ should satisfy, while $\Delta T_{hyst}$ serves as a lagging parameter to avoid oscillation of the optimization results, ensuring the density of the entire optimization results. The trajectory nodes are associated with obstacles and waypoints near the trajectory, and each trajectory point corresponds to the respective obstacles. The algorithm transforms the optimization problem into a hyper-graph, which is different from a basic two-dimensional graph. In this hyper-graph, the nodes connected by an edge are unrestricted, and the constraints are the edges of the hyper-graph. The optimization variables related to it are the nodes connected by the edge. This transformation allows constraints to act only on local optimization variables, such as the velocity constraint in Figure 6, which is only related to the front and back poses of the vehicle, $s_{0}$, $s_{1}$, and their time interval $\Delta T_{0}$. Therefore, the hyper-graph has three nodes and one edge. Thus, the entire optimization problem can be represented as a sparse matrix, and the open-source framework *g2o* is used to solve this optimization problem. Once the optimal trajectory $B^{*}(Q,\tau )$ is obtained, on the one hand, using Equation (8), the required motor speed $v_{control}$ and equivalent front wheel steering angle $\delta_{control}$ for the wheel controller are calculated. On the other hand, leveraging the algorithm framework, with the movement of the vehicle, after inserting and deleting states, re-optimization is carried out. This iterative optimization strategy greatly improves computational efficiency and ensures trajectory quality.

In contrast to directly selecting the minimum motion cost, we use the total time of the trajectory as the objective function to improve exploration efficiency:(15)$$min\sum_{k=1}^{n-1}\Delta T_{k}^{2}.$$

By combining the above constraints and objective functions, we transform the nonlinear programming problem with hard constraints into a quadratic optimization problem to improve the efficiency of solving. Simultaneously, we utilize the sparsity of the problem and transform constraints into penalty terms. For the equality constraint in Equation (9), it is transformed into quadratic constraints, as follows:(16)$$\phi (h_{k},\sigma_{h})=\sigma_{h}{h^{T}}_{k}Ih_{k}=\sigma_{h}{{\left\|h_{k}\right\|}^{2}}_{2},$$
where I is the identity matrix, and $\sigma_{h}$ is the penalty value for this constraint.

For inequality constraints $\upsilon_{k}$ such as (10), (11), (12), (13), they can be converted into one-sided quadratic constraints with penalty value $\sigma_{k}$:(17)$$\chi (\upsilon_{k},\sigma_{v})=\sigma_{v}{{\left\|min\{0,\upsilon_{k}\}\right\|}^{2}}_{2}.$$

We transform the final objective function into the following form:(18)$$\begin{array}{l} \;B_{*}=\underset{B}{min}\tilde{V}(B)\\ \tilde{V}(B)=\begin{array}{l} \sum_{k-1}^{n-1}\left[\Delta {T_{k}}^{2}+\phi (h_{k},\sigma_{h})+\chi (\rho_{k},\sigma_{\rho })+\chi (\nu_{k},\sigma_{v}\right)\\ +\chi (\alpha_{k},\sigma_{\alpha })+\chi (o_{k},\sigma_{o})]+\chi (\alpha_{n},\sigma_{\alpha }) \end{array} \end{array}.$$

Here, ΔT represents the time interval between two poses, $h_{k}$ is the kinematic constraint, $\rho_{k}$ is the curvature constraint, $\nu_{k}$ and $\alpha_{k}$ are the velocity and acceleration constraints, $o_{k}$ represents the distance constraint from obstacles, and $\sigma_{h}$, $\sigma_{\rho }$, $\sigma_{v}$, $\sigma_{\alpha }$, and $\sigma_{o}$ are penalty matrices. This structure provides convenience for adding new constraint conditions in the future.

The overall algorithm flow is shown in Figure 7. Firstly, based on the global reference path obtained from solving the TSP, the initial path is interpolated. Dynamically insert or delete optimized state variables according to the vehicle’s motion, then match the trajectory nodes with the obstacles and waypoints near the trajectory, find the corresponding obstacle for each trajectory point, establish a hyper-graph using the objective function in Equation (18), and optimize to obtain the optimal trajectory $B^{*}(Q,\tau )$; when the trajectory is feasible, calculate the vehicle’s control inputs based on Equation (8) and send them to the lower-level controller.

## 5. Experiments and Discussion

In this section, we use the Gazebo simulation software (Gazebo 11.0, accessed on 12 November 2023) to construct three different enclosed environments of various sizes to evaluate the feasibility of our algorithm. As shown in Figure 8, the algorithm is based on ROS kinetic (ROS1 kinetic, accessed on 12 November 2023) under Ubuntu 16.04, which is implemented using the C++ programming language. Gazebo is used to model the environment, and physical models of the vehicle’s motor, steering system, LiDAR sensor, and others are created to build an Ackermann-based unmanned ground vehicle model. The RVIZ component of ROS receives topics such as robot status and sensor data from Gazebo for display. The computer used in the simulation experiment is equipped with an Intel^®^ Core^TM^ i7-9750H @ 2.60 GHz (Intel Corporation, Santa Clara, CA, USA) and a Nvidia GeForce GTX 1060 4 GB (NVIDIA Corporation, Santa Clara, CA, USA).

We use a 0.4 m × 0.2 m Ackermann-steered four-wheel small car with 360° Lidar as the vehicle model for the experiment. The parameters of the unmanned vehicle system are shown in Table 1. Considering the real situation, the perception length of the vehicle is set to about ten times the length of the vehicle, approximately 5.5 m. The minimum turning radius, maximum speed, maximum angular velocity, and maximum acceleration of the vehicle are 0.8 m, 2 m/s, 0.6 rad/s, and 0.5 m/s^2^, respectively. For our algorithm, considering the completeness of probability when constructing the PRM graph and the size of the actual simulation environment, combined with multiple debugging results, we set the upper limits for nodes and edges to 1500 and 500, respectively. We set the step size for expanding nodes $d_{grow}$ to 1 m, the distance for determining connected edges $d_{near}$ to 0.5 m, and the threshold for filtering boundary points $gain_{threshold}$ to 10.

Efficient autonomous exploration requires correct goal points or global path guidance and smooth executable local trajectories. The choice of target point or global path determines the robot exploration efficiency, motion continuity, and global optimality, and is crucial for autonomous exploration. As presented in Section 2, RH-NBV [9] and GB [12] proposed different strategies for the exploration guidance problem and achieved better results. Unlike them, we propose a combination of incremental B-splines and probabilistic roadmaps to improve the quality of frontiers and global paths, while we improve the traditional TEB algorithm for the kinematics and dynamics constraints of UGVs to serve as our local planner to smooth the global paths so that they can be executed by UGVs. In view of this, we choose two mainstream exploration methods, RH-NBV and GB, as comparative experiments to verify the improvement of exploration efficiency and continuity of our method. Since the local planners of the RH-NBV and GB algorithms are only applicable to disc robots or Turtlebots that satisfy the integrity constraints, to better carry out the comparison experiments, we also use TEB as the local planning component of the two comparison algorithms to plan smooth trajectories suitable for the execution of UGVs, and all the three algorithms use gmapping as the graph building component.

### 5.1. Scenario I Exploration Experiment

The first scenario, as shown in Figure 9a, belongs to a typical binary intersection structure, with an area of about 50 m^2^. This structure can fully verify the exploration and backtracking capabilities of the unmanned vehicle, that is, to verify whether the vehicle can find the next boundary point outside the local area and make an effective transition after completing the exploration of the local area. Figure 9b clearly shows the motion trajectories after the completion of the three algorithms’ explorations, with the red part representing our algorithm. Compared to the trajectories of the GB algorithm in green and the RH-NBV algorithm in blue, the trajectory of our algorithm is smoother, with better curvature continuity, and more conducive to continuous acceleration of the vehicle.

After the exploration of three algorithms in Scenario I, the data of motion distance and time are shown in the table below.

From Table 2, it can be seen that the total length of the exploration paths of the three algorithms is relatively small, all of which have backtracking functionality. However, the algorithm proposed in this paper has a significantly shorter total exploration time compared to the other two algorithms, showing an improvement of nearly 32.5% and 17.8% over the GB algorithm and the RH-NBV algorithm, respectively. This is because the map is relatively small, and the distances covered by the three algorithms are similar. The algorithm proposed in this paper utilizes B-spline curves to search for paths, considering the current speed orientation of the vehicle. Moreover, it optimizes the path towards the target point using a rolling optimization framework within the algorithm. Therefore, during the motion process, especially when switching frontiers, the vehicle does not stop due to discontinuous paths. Additionally, the path generated has fewer sharp corners, making it smoother and more conducive for the vehicle to explore at higher speeds, thereby improving overall efficiency.

### 5.2. Scenario II Exploration Experiment

As shown in Figure 10a, in Scenario II, a small enclosed environment with obstacle cubes is set up, which is an upgrade of Scenario I. Similarly, it requires exploration of various small areas, followed by backtracking or other processes after completing local exploration. However, the intersection structure is more complex, requiring more obstacle avoidance processing. The trajectory results in this area are shown in Figure 10b.

From the above figure, it can be noted that compared to the other two algorithms, our algorithm constructs the full map in a very smooth manner. In contrast, the trajectories generated by our algorithm do not have sharp corners, indicating that the vehicle does not need to stop to change directions while moving. The exploration distance and time data are shown in the Table 3 below:

It can be seen that for maps with multiple forks, our algorithm, considering global optimality, has a significant advantage, with both total path length and time being noticeably better than other algorithms. In terms of total path length, there is an improvement of about 52.9% compared to the GB algorithm and about 64% compared to the RH-NBV algorithm; in terms of exploration time, our algorithm has increased by approximately 67.9% compared to the GB algorithm and about 75.9% compared to the RH-NBV algorithm. The paths generated by our algorithm are smoother without sharp corners. In addition, when the other two algorithms encounter forked intersections, they tend to sway left and right due to similar distances and information gain values between the two areas. Figure 11 illustrates that the GB algorithm generated two global paths (red straight lines) in opposite directions at the same location during operation, causing the vehicle to oscillate between left and right, potentially leading to exploration failure due to the inability to break the balance. However, our algorithm, by adding B-spline control points with motion orientation in the node attributes, selects different frontiers with different motion costs, fundamentally avoiding such issues.

### 5.3. Scenario III Exploration Experiment

As shown in Figure 12a, Scenario III is a simulation of a large complex indoor environment, covering an area of approximately 100 square meters. There are more intersections and numerous obstructions, making the structure quite complex. In this scenario, the trajectories obtained using our algorithm are shown in Figure 12b. Similarly, the trajectories planned by our algorithm exhibit good curvature continuity, making them smoother compared to other algorithms.

The exploration distance and total time for Scenario III are shown in Table 4:

It is evident that both the time and distance of our algorithm are better than the other two algorithms. In terms of total exploration path length, we have improved by approximately 26.4% compared to the GB algorithm and about 29.3% compared to the RH-NBV algorithm; in terms of exploration time, our algorithm has increased by about 53.7% compared to the GB algorithm and around 61% compared to the RH-NBV algorithm. The trajectories generated by our algorithm are smoother and able to avoid local optima. Figure 13 demonstrates that the GB algorithm, due to not considering the current orientation and posture constraints of the vehicle when calculating the global path, outputted a locally optimal global path to the TEB planner, causing it to become stuck at the corners and resulting in a delayed exploration.

On the other hand, in exploration, the left and right swaying caused by the equal weights of frontiers is more pronounced. Taking the RH-NBV algorithm as an example, as shown in Figure 14, the vehicle, during its motion, becomes stuck in a situation of continuous left and right swaying due to the close weight values of the two nodes.

In summary, for autonomous vehicles, our algorithm outperforms the RH-NBV and GB algorithms in terms of distance and time. In environments with small exploration areas, as shown in Scenarios I and II, the trajectories are smoother and the motion speed is faster. In the large area map depicted in Scenario III, by using the TSP to solve for optimal frontiers, our algorithm covers a shorter total distance and achieves higher exploration efficiency.

## 6. Conclusions

In this paper, we propose a hierarchical planning method that combines incremental B-splines with a probabilistic roadmap to enable autonomous vehicles to conduct fast and efficient exploration in complex unknown environments. To improve the efficiency and continuity of vehicle motion during exploration, we define a new boundary point structure that provides guidance in unknown spaces and includes curve segments that are more suitable for vehicle motion characteristics. By constructing and maintaining global and local graphs to store frontiers and generate optimal global paths, we ensure smooth and constraint-satisfying feasible trajectories through further optimization using the time elastic band algorithm. Comparative experiments with mainstream advanced exploration methods in simulated environments of different scenarios demonstrate that our approach effectively enhances the exploration efficiency of autonomous vehicles.

## Figures and Tables

**Figure 1 sensors-24-03951-f001:**
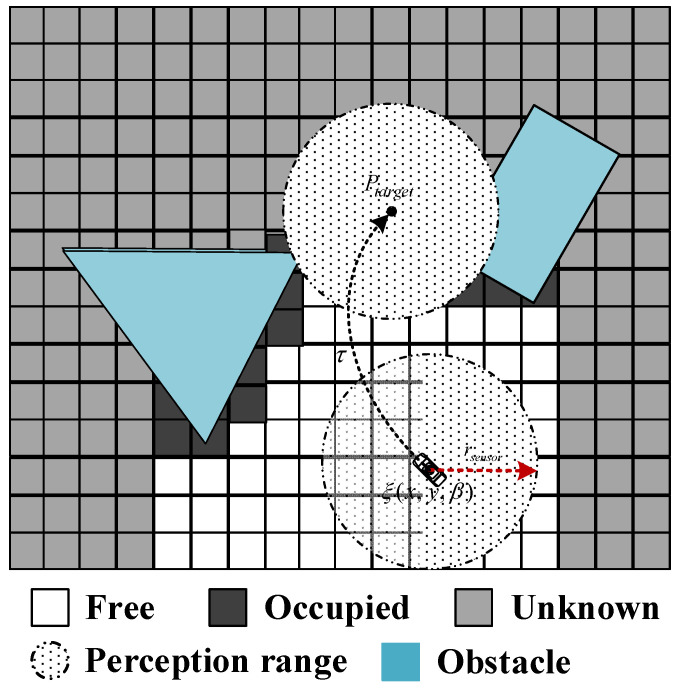
Explore the map and the information gain schematic.

**Figure 2 sensors-24-03951-f002:**
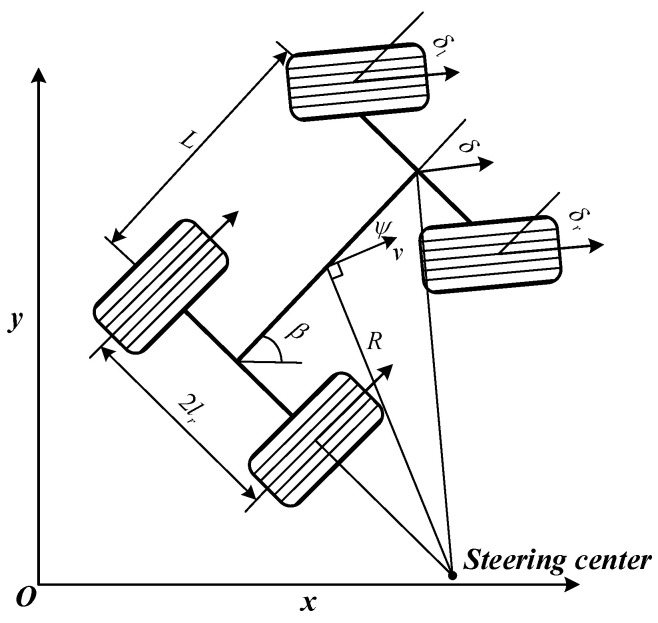
Kinematic model of vehicle.

**Figure 3 sensors-24-03951-f003:**
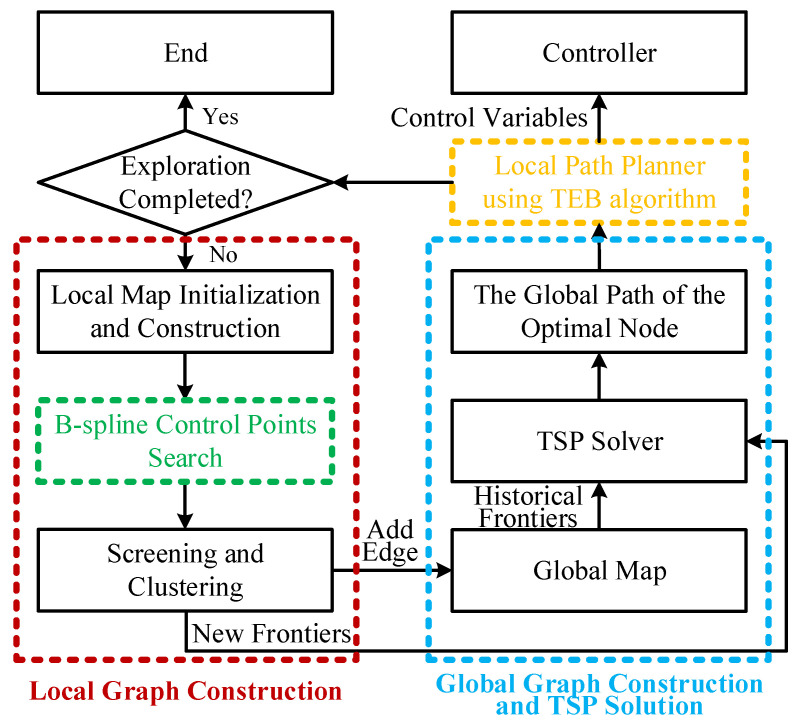
An overview of the proposed exploration framework.

**Figure 4 sensors-24-03951-f004:**
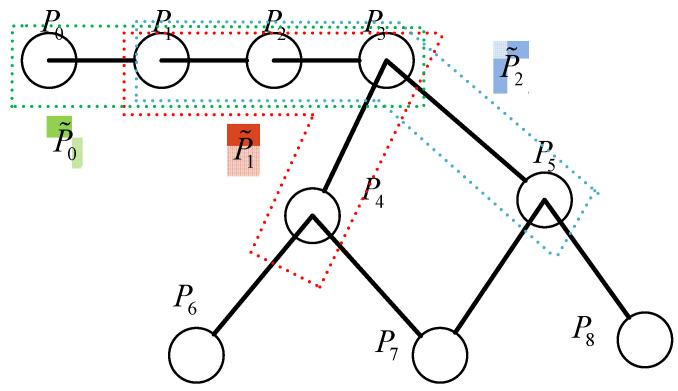
Schematic representation of the relationship between the new map and the original map.

**Figure 5 sensors-24-03951-f005:**
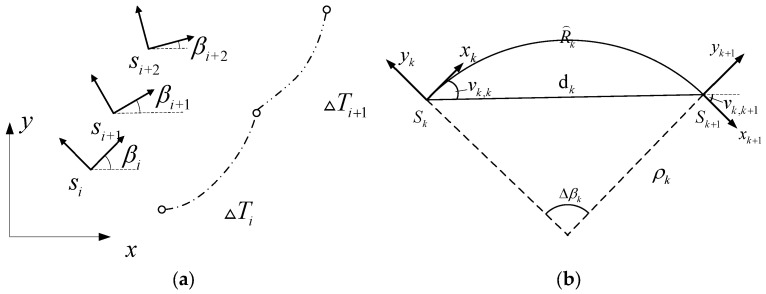
Time elastic band parameter schematic: (**a**) UGV pose time sequence diagram, and (**b**) curve path schematic.

**Figure 6 sensors-24-03951-f006:**
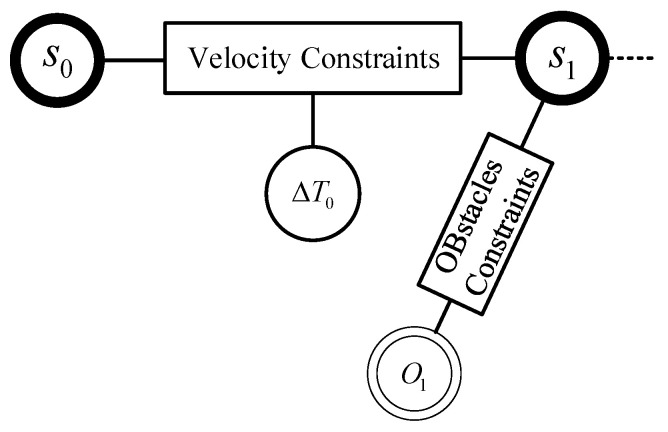
Hyper-graph structural schematic.

**Figure 7 sensors-24-03951-f007:**
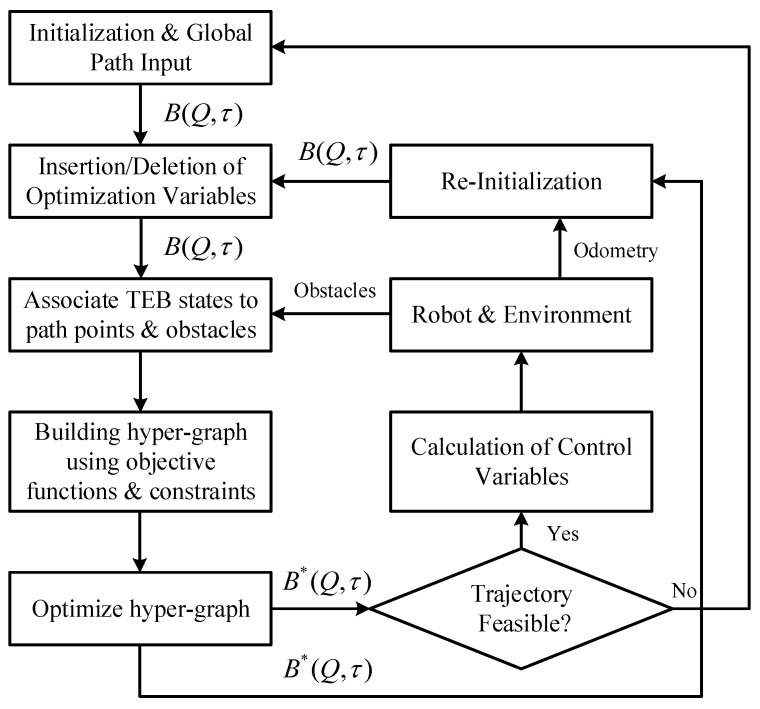
Flowchart of local trajectory planning algorithm.

**Figure 8 sensors-24-03951-f008:**
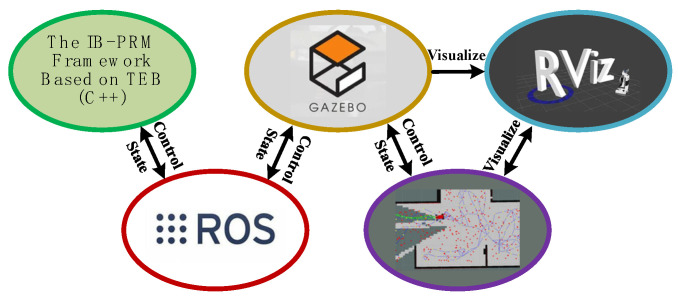
The simulation experiment process of proposed IB-PRM framework.

**Figure 9 sensors-24-03951-f009:**
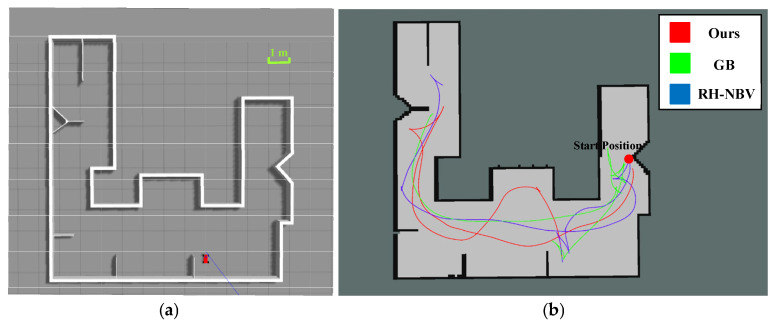
(**a**) Top view of Scenario I and (**b**) trajectory display after three algorithmic explorations are completed.

**Figure 10 sensors-24-03951-f010:**
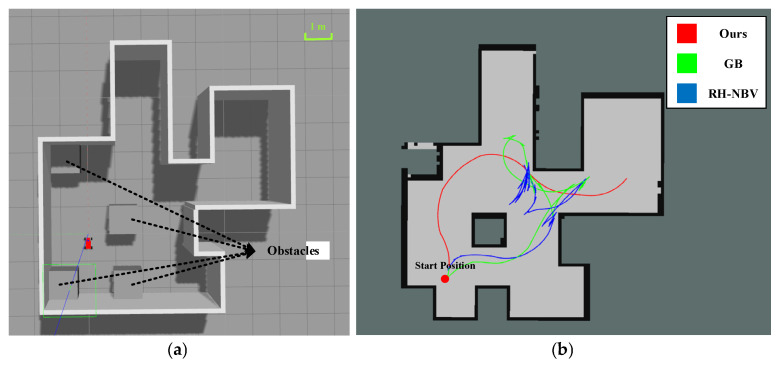
(**a**) Top view of Scenario II and (**b**) trajectory display after three algorithmic explorations are completed.

**Figure 11 sensors-24-03951-f011:**
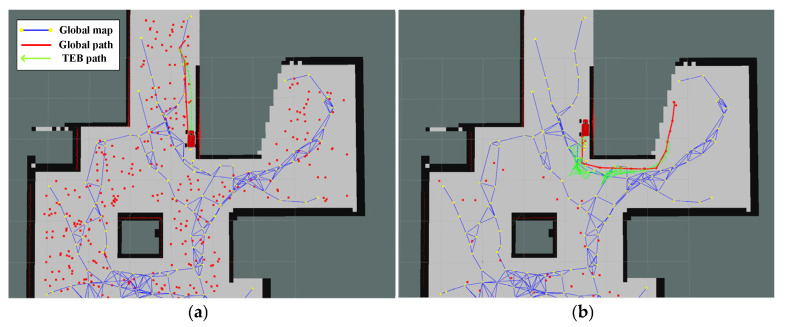
During the operation of the GB algorithm, it falls into a situation of left–right oscillation. The red scattered points in the figure represent the nodes expanded by the GB algorithm when constructing the PRM graph. (**a**) The GB algorithm plots a leftward path at a location and (**b**) the GB algorithm plans a rightward path at the same location.

**Figure 12 sensors-24-03951-f012:**
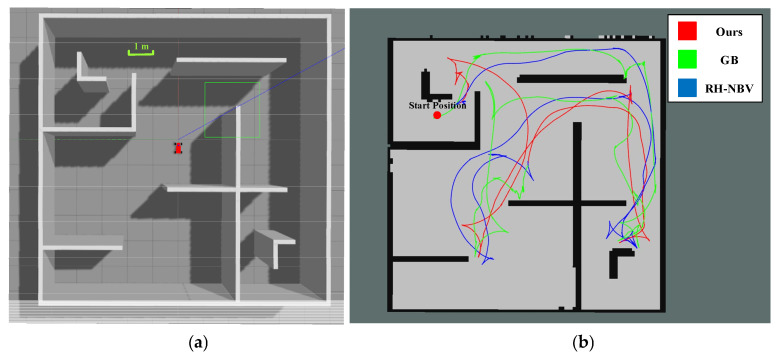
(**a**) Top view of Scenario III and (**b**) trajectory display after three algorithmic explorations are completed.

**Figure 13 sensors-24-03951-f013:**
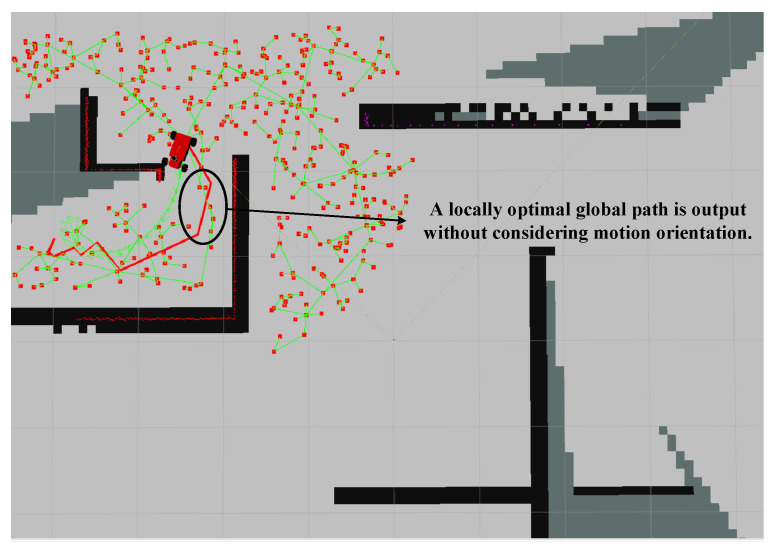
The case where the vehicle is stuck due to the GB algorithm outputting a locally optimal global path. The red scattered points in the figure represent the nodes expanded by the GB algorithm when constructing the PRM graph. The green lines represent the edges of the local PRM map. Lines in fluorescent color with arrows form TEB paths.

**Figure 14 sensors-24-03951-f014:**
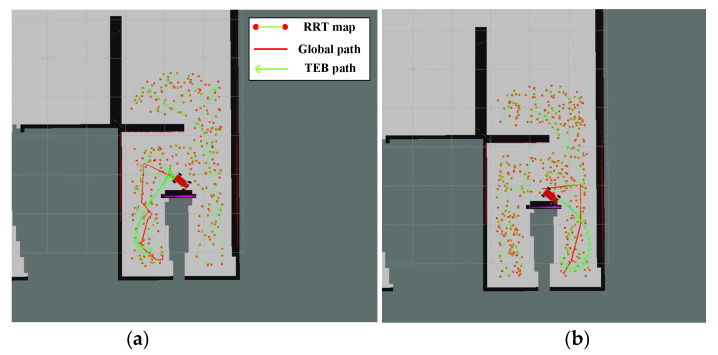
The RH-NBV algorithm falls into a wobble when the left and right boundary points have the same weight. The red scatter in the figure shows the nodes of the tree when the RH-NBV algorithm is building the RRT tree. (**a**) The RH-NBV algorithm plots a leftward path at a location and (**b**) the RH-NBV algorithm plans a rightward path at the same location.

**Table 1 sensors-24-03951-t001:** Parameters related to unmanned vehicles and systems.

Experimental Parameter	Symbolic	Setpoint
** *Perception range* **	$r_{sensor}$	5.5 m
** *Max nodes* **	$Node_{max}$	1500
** *Max edges* **	$Eage_{max}$	500
** *Extension step size* **	$d_{grow}$	1 m
** *Determine step size* **	$d_{near}$	0.5 m
** *Gain thres.* **	$gain_{threshold}$	10
** *Min turning rad.* **	$R_{min}$	0.8 m
** *Max vel.* **	$v_{max}$	2 m/s
** *Max angular vel.* **	$\omega_{max}$	0.6 rad/s
** *Max acc.* **	$a_{max}$	0.5 m/s^2^

**Table 2 sensors-24-03951-t002:** Scenario I explores data comparison.

Scenario I	GB	RH-NBV	Ours
Total path length (m)	32.173	31.5202	33.706
Exploration time (s)	89.6435	71.262	60.503

**Table 3 sensors-24-03951-t003:** Scenario II explores data comparison.

Scenario II	GB	RH-NBV	Ours
Total path length (m)	22.365	28.996	10.43
Exploration time (s)	71.926	95.804	23.051

**Table 4 sensors-24-03951-t004:** Scenario III explores data comparison.

Scenario III	GB	RH-NBV	Ours
Total path length (m)	58.8729	61.2843	43.2884
Exploration time (s)	282.548	303.221	130.74

## Data Availability

Data are contained within the article.

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
