# Peer review of "Autonomous Exploration Method of Unmanned Ground Vehicles Based on an Incremental B-Spline Probability Roadmap"

_sensors, 2024, doi:10.3390/s24123951_

Round 1
Reviewer 1 Report
Comments and Suggestions for Authors
This paper proposes a new approach of the path planning in unknown environments. However, I'm not sure the novelty of this research due to difficult readability of this paper while I'm not familiar with the robot path planning.
Followings are my major comments.
[1]The readabilty of this paper is not so good. One of the reasons is usage of many symbols, and it seems that sufficient explanation of each paramter is not provided. This will confuse the readers.
For example, you use the symbols of ξ and s to represent the pose of vehicle. What are different these parameters? It is better to organize the used symbols.
(For example, you can create a parameter list in Appendix.)
[2]It seems that writing of [V Proposed Approach ] is not organized well. I thik that this chapter is important. However, due to diffucility of reading of this chapter, I'm not sure the novelty of your method.
[3]I think that Eq.(18) and Fig.7 are important for your method because they may indicate total function and flow.However,It is not clear that which compornents in the equation and figure corresponds to each section in Chapter V.
I suggest that you should firstly indicate Eq.(18) and Fig.7 and novelty of your method. Then, you can explain each component of the flow.
[4]You should explain an assumption of the onboard sensor. You describe that the sensor perception range is 5 m in circle shape. Do you assume cheap 360deg LIDAR as onboard sensor? If you assume only the visible camera as onboard sensor, does the perception range change?
[5]It is better to add some explantions about parameter design of the expriment listed in Table1. It will be helpful to understand whether these designs are appropriate.
[6]You should add start and goal positions, location of obstacle and the map scale in the maps shown in Figs.8,9.11.
[7]In your experiment, you compare the results derived from GB and RH-NBV with those from yours. However, I'm not sure whether these comparisons are appropriate.
I think that key points of your method are utilization of B-spline curves and consideration of robot's kinematic constraints.
Aren't there other path planning researches using B-splin curves and/or kinematic constrainsts?
[8]In Fig.10 and 12, there are many red points in the map. What are they? You should add more explanations about each figure.
[9]Please confirm the fromat of your paper. It seems that format of your paper dparts from the format of [Sensors]. You should fit the format to that of the [Sensors].
[10]Please confirm whether citation of each reference is correct. For example, I may not find citation of reference [1].
Comments on the Quality of English Language
I think you need moderate correction of English.
Reviewer 2 Report
Comments and Suggestions for Authors
This manuscript presents a hierarchical planning method that combines Incremental B-splines with Probabilistic Roadmaps. The manuscript still needs further revisions. The detailed comments are given below:
1. Abstracts need to be carefully revised.
2. The introduction should be well organized to emphasize the novelty of this paper. It is not clear to read. The state-of-the-art should be improved in the Introduction.
3. The abstract section needs to summarize the current state of research and should not be a list of literature.
4. The contribution part needs to be further described, such as the improvement of the methods.
5. The experiment is only roughly presented. Please include more details on the experimental setup, experimental environment, specifications of the test object, test procedures, data processing, and parameter selection.
6. The formatting of references is not uniform and needs to be fixed.
7. The format is very cluttered, e.g. table1, table2.
8. Layout of articles needs to be scrutinized
Comments on the Quality of English LanguageNo comments
Round 2
Reviewer 1 Report
Comments and Suggestions for Authors
Thank you for revising your manuscript according to my comments. I think that readabilty of this manuscript is improved. However, some improvements are still required before publication.
Followings are my additional comments.
[1]I found that your main novelty is B-spline control point algorithm and maybe TEB algorithm in local path planner.
On the other hand, you say that GB algorithm and RH-NBV algorithm used in your experiment also adopt TEB alogorithm. Are main differences between your method and their methods usage of B-spline algorithm?
I think that it is better to clarify what components enable smother path and more rapid exploring time in your algorithm compared the to previous methods. This is my major comment.
[2] In Eq.(13), you use symbols [s1],[O1] repeatedly. Are they correct?
It seems that [s1],[O1]→[s2],[O2]... are better. Then, in line 438, where is [δmin] in Eq.(13)? And, in line 484, is B*(Q,τ) B*(s,τ)?
Anyway, I think that you had better check whether used symbols and notations are correct throughout whole paper again.
[3] In Eq.(14), are the directions of [>] correct? I think that the direction is contrary. If it is correct, it's OK. But, I think that you should also check each equation and its notation.
[4]Thank you for adding the parameter list in Appendix. You had better write that main parameters used in this research are listed in Appendix anywhere in the main text.
[5] Adding of Fig.3 (overview) is also good idea. I think that it is better to add the discription of [TEB algorithm] in yellow dashed line such as [Local Path Planner using TEB algorithm].
Then, the words of [Add Edge], [New Frontiers] are hidden under the figure, they can be also modified. They are my minor suggestions.
[6] You have corrected for correspondence between the reference list and the citations in the main text. But, in line 242, Usenko [17] is referred in [18] in the reference list. Please check the correpodences including other citations again.
It is better to perfom minor check including the symbols notations.
Reviewer 2 Report
Comments and Suggestions for Authors
I have no comments.
Comments on the Quality of English LanguageI have no comments.
Author Response
Reviewer2 have no comments.
Round 3
Reviewer 1 Report
Comments and Suggestions for Authors
Thank you for revising your manuscript according to my comments. I think that your paper can be published after final check and correction as may be necessary.
Please refere following comments.
[1]Do you put the reference numbers automatically in main text? I think that reference numbers in the main text and those in the reference list are not consistent yet. Maybe, they are off by one.
[2]The abbreviations such as UGA, TEB should be written as full terms like Unmanned Ground Vechicles when they are firstly appear in the main text independently of Abstract.
Anyway, I recommend you to perform final checks of descriptions, symbols, equations and figures. After that, if the editors and another refree approve your paper, please publish this paper.
Comments on the Quality of English Language
I think that English quality reaches the publication level after minor check of their discriptions
